# *Gemmamyces piceae* Bud Blight Damage in Norway Spruce (*Picea abies*) and Colorado Blue Spruce (*Picea pungens*) Forest Stands

Michal Samek [1], Roman Modlinger [1,*], Daniel Baťa [1], František Lorenc [2], Jana Vachová [1], Ivana Tomášková [1] and Vítězslava Pešková [1]

1 Faculty of Forestry and Wood Sciences, Czech University of Life Sciences Prague, Kamýcká 129, 6-Suchdol, CZ-165 21 Prague, Czech Republic; samek@fld.czu.cz (M.S.); bata@fld.czu.cz (D.B.); ruckova@fld.czu.cz (J.V.); tomaskova@fld.czu.cz (I.T.); peskovav@fld.czu.cz (V.P.)

2 Forestry and Game Management Research Institute, Strnady 136, CZ-252 02 Jíloviště, Czech Republic; lorenc@vulhm.cz

* Correspondence: modlinger@fld.czu.cz; Tel.: +420-606-688-883

**Abstract:** Since 2008, spruce bud blight (*Gemmamyces piceae* (Borthw.) Casagr.) has been spreading epidemically in forest stands of the Czech Republic's Ore Mountains. This fungus, with a disjunct Holarctic range, injures buds, especially of Colorado blue spruce (*Picea pungens* Engelm.). Damaged buds do not sprout, and, in case of a stronger attack, the tree does not recover its assimilation apparatus and may die. Within the past few years, there has been a huge spread of this fungus throughout the mountain range. This paper summarizes the biology of *G. piceae*, its host plants, and presents the first findings from the massive outbreak of *G. piceae*. In 2015, an increase in damage was detected on Norway spruce (*Picea abies* (L.) Karst). The course of the *G. piceae* epidemic had been monitored in 25 permanent research plots over the course of 11 years. In the case of Colorado blue spruce, stands aged 10–60 years were attacked, with 60% of buds damaged on average. The intensity of damage to Norway spruce buds was around 25%. Norway spruce infestation varied significantly depending upon the age of the stand (GLMM: $p < 0.01$). In the age class of 31–60 years, on average 80% of individuals were infested. In older stands, only 42% of trees were infested, and no infestation was found in individuals younger than 15 years. In Colorado blue spruce, the distribution of the pathogen was continuous, whereby all individuals in the research plots were affected, and, with the exception of a few trees, the infestation was lethal or resulted in a significant reduction of the assimilation apparatus. The development of damage on Colorado blue spruce can be characterized as continuous growth.

**Keywords:** forest pest; damage; forest health; disease; forest pathology; fungi; invasive species

## 1. Introduction

Air pollutants such as ozone (O₃), nitrogen oxides (NOₓ), and sulfur dioxides (SOₓ) contribute significantly to deteriorating forest health [1]. In Europe, this phenomenon has been observed in many countries, such as in Germany, Poland, Austria, or Switzerland [2], and, in Central Europe, it led to the destabilization and collapse of forest stands in many places [3,4]. In the Ore Mountains region, problems with the health of forest stands were recorded as early as the end of the 16th century [5], and in the case of air pollution from the beginning of the 20th century [6]. The first notable damage to forest stands is related to the beginning of coal burning [7]. In the 1980s, after an extreme air pollution load, most of the spruce and beech forests collapsed. In areas where it was not possible to restore these trees, stands of substitute tree species were established [8]. These were species with potentially higher air pollution resistance, and among conifers, mainly Colorado blue spruce. Until the beginning of the 21st century, the selection of this tree species seemed to be

very appropriate. Forest stands of Colorado blue spruce in the Ore Mountains covered an area of more than 8000 ha [9]. Individual trees were already reaching dimensions suitable for the timber industry, and plans for economic exploitation of this tree species gradually began to emerge. Improving the health of forest stands was also facilitated by reducing the air pollution. However, significant soil acidification as a consequence of atmospheric pollution persisted, making vital forest stands highly unstable [10]. Even after 30 years, the fungal and microbial communities living in the soil failed to recover [11]. Nevertheless, even the reduced ectomycorrhizal fungal diversity is still beneficial to the host tree [12]. Next to the increased water and mineral uptake by mycorrhizae fungus, the advantage of the mycorrhizal presence are the fungal sugar products of the metabolism. According to our previous research, the content of trehalose is significantly higher in healthy trees compared to the trees with bud blind disease [13].

After a long period without a record of its occurrence in the Czech Republic, *G. piceae* was detected during 2008 on Colorado blue spruce in the Ore Mountains [14]. Initially rather inconspicuous and localized, the damage started to take on an epidemic character in 2009, and especially in the northeastern part of the Ore Mountains region, reaching disastrous proportions within a short period of time [15,16].

The fungus *G. piceae*, synonym *Cucurbitaria piceae* Borthw., *Cucurbidothis piceae* (Borthw.) Petr. was first found on infected buds of Colorado blue spruce (*P. pungens* var. *glauca*) at Abercairney, Perthshire (U.K.) in 1906 [17,18]. Under current taxonomic nomenclature, it is classified in the division Ascomycota, order Pleosporales, family Melanommatacea [19]. The anamorphic stage described by Naumov in 1925 is referred to as *Megaloseptoria mirabilis* Naumov [20].

Infection of the host plant occurs during the vegetation season. The following year, at the time of budding, first pycnidia form, and conidia then form within them. Conidia occur within the fruiting bodies until September, with strong production during July and August [21]. This is followed by the formation of perithecia, where ascospores develop and form from April to August, with strong production in the summer months (July and August). The peak of ascospore production is slightly delayed after the peak occurrence of conidia. Both types of fruiting bodies can be encountered on buds during the vegetation season. The infestation of an individual causes buds swelling and spiral twisting. When the infection is severe, and the terminal bud dies, a more pronounced growth of lateral shoots is typical. In spring, infected buds are covered with a hard, black crust (basal stroma of the fungus), from which numerous small brown- to black-coloured spherical fruiting bodies grow, or the fruiting bodies grow directly between the scales of the bud, and, in case of a stronger infestation, they cover the entire bud [22]. The presence of buds with bud-blight fruiting bodies does not immediately threaten the infested tree. If the percentage of infested (and dead) buds is relatively low, the tree's growth is not significantly restricted. If the number of infected and dead buds exceeds three-quarters of the total number for several years in a row, however, the tree stops budding, does not renew its assimilation apparatus, and may die.

The pathogen has been confirmed in a number of European countries, such as, Austria, Denmark, England, Finland, Germany, Ireland, Italy, Russia, Scotland, Slovenia, Sweden, Switzerland and Wales [17,18,23–32]. In the Czech Republic, *G. piceae* was first detected on Colorado blue spruce in 1917, but damage showing the same symptoms was observed as early as 1909 [23]. From 1910 onwards, the fungus was repeatedly found in many parts of Bohemia [32]. With regard to the frequency of the pathogen's findings after 1910 [23,33], Central Europe was considered to be the epicenter of its spread. Thereafter, though, the fungus was not detected in the Czech Republic until the 21st century.

The infection was described as not very significant. The most severe damage was recorded in northern England and southwest Scotland, where damage to older individuals of Norway spruce was found in 13 plots [34]. In addition to Europe, the pathogen has been found in North America, in Alaska [35].

The spruce bud blight primarily damages North American spruces Colorado blue spruce and Engelmann Spruce (*Picea engelmannii* Parry ex Engelm.), which are usually regarded as susceptible [36]. In Alaska, its occurrence has been confirmed on White spruce (*Picea glauca* Moench), Black spruce (*Picea mariana* (Mill.) Britton, Sterns and Poggenb.), and Sitka spruce (*Picea sitchensis* Bong.) [35]. This applies also to the Czech Republic, where the pathogen has been confirmed on *P. glauca*, but recently also on the south European *Picea omorika* (Pančić) Purk. [37]. Systematic monitoring of the wide spread of the pathogen on Colorado blue spruce in the Czech Republic was carried out by Pešková et Soukup [38], Pešková et Modlinger [37], Černý et al. [20] and Šefl et al. [39].

Due to the considerable extent of damage to the Colorado blue spruce stands in the Ore Mountains, resulting from *G. piceae* infestation, the stands could be regarded as highly destabilized, and it was necessary to proceed with stand reconstruction. Moreover, an increase in the number of pathogen detections on Norway spruce in 2015 began to raise concerns about further development of these stands, as the infectious capabilities of the fungus was strong [40].

The main objective of this research was to compare the development of bud-blight infestation on the two host tree species, Colorado blue spruce and Norway spruce, in terms of the intensity of damage to individual trees and spatial extent of the infestation in the monitored stand. The results presented here are based upon 11 years of research in the Ore Mountains region, which is currently the epicenter of the bud blight's occurrence. Findings from the current widespread outbreak act as an essential information source for areas that have been potentially newly affected by this pathogen, such as Alaska or other European countries. Secondary objectives were to summarize current knowledge about the distribution of *G. piceae* in the world, list the host tree species known to date, and describe the biology of the pathogen and factors influencing the spread of the fungus in stands.

## 2. Materials and Methods

The decline of replacement tree stands in the Ore Mountain region was monitored between 2009 and 2020. In 2009, research plots with Colorado blue spruce were established. These were homogeneous stands of the same age, where 25 individuals were selected and repeatedly evaluated. Due to the intensive spread of spruce bud blight on Norway spruce, research plots were established in 2016 also in homogeneous stands of this tree species. In each plot, 30 individuals were monitored. To compare Colorado blue spruce and Norway spruce, only research plots from the location around the Fláje waterworks were selected (Figure 1), as this was the epicenter of the disastrous occurrence and data on the intensity of infestation were found to be most complete there.

To monitor the intensity of spruce bud blight infestation, we followed the methodology of Černý et al. [20] using a scale from 0 to 4, where 0 = trees without damage, 1 = trees with low damage (up to 25% of buds affected), 2 = moderate damage ($25\% \leq x < 50\%$ of buds affected), 3 = high damage ($50\% \leq x < 75\%$), and 4 = extreme damage ($\geq 75\%$ of buds affected). Trees in the last category were dying or dead. The values of disease severity in particular trees were averaged for each plot. The resulting values were linearly approximated as percentages based upon marginal values for each category (0%, 25%, 50%, 75%, and 100%).

For further processing, two variables were created to express the infestation by spruce bud blight, namely, the proportion of damaged buds and proportion of attacked trees. Proportion of damaged buds is the average rate of attacked tree canopies calculated from all individuals within a plot as a percentage, and thus it expresses the intensity of damage to individual trees. The proportion of attacked trees is the proportion of attacked individuals among all individuals in the evaluated plot, and thus it is an indication as to the spatial extent of infestation in the studied stand. A direct comparison between Colorado blue spruce and Norway spruce could be made only in the age class of 31–60 years, as Colorado blue spruce has only been planted in the Ore Mountains since the 1970s [9].

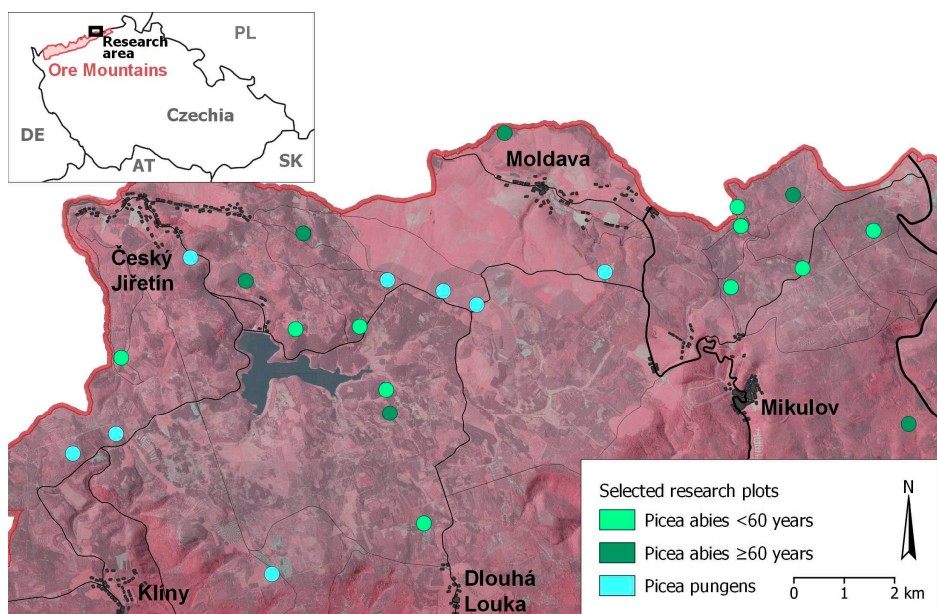

**Figure 1.** Location of research plots in the Czechia and the Ore Mountains. The colours of the point represent different tree species or age classes of forest stands.

For the statistical evaluation, we created a dependent variable which is the percentage of infested individuals in the stand (sum of individuals from categories 1–4 for a given stand divided by the number of evaluated individuals in the stand). The year of evaluation was set as a categorical dependent variable and age was set as a discrete dependent variable. Basic assumptions for the regression-type statistical analysis were made following Zuur et al. [41]. The dependent variable was not well approximated by a normal distribution, and the Gamma distribution function was shown to be the most appropriate. Moreover, the assumption of independence was violated from the viewpoint of evaluating the same stand in all years. This circumstance was incorporated into the statistic model by using stand as a nested factor. A generalized linear mixed effects model in the glmmTMB package was used to evaluate the data under the procedures described by Brooks et al. [42]. All statistical analyses were performed in the R 4.0.2. environment [43].

## 3. Results

The pathogen developed very rapidly on Colorado blue spruce. In 2009, an average of 55% of the buds were infected, by 2010 this percentage reached 63%, and thereafter the damage fluctuated at around 60% of infected buds. In 2015, there was a further increase in infestation to 65% (Figure 2). Infestation at this level is already a very serious problem for the tree, and the chances for successful regeneration of the assimilation apparatus are restricted. The relatively strong bud damage in 2009 indicates that the pathogen must have been already present in stands in the previous period. Nevertheless, the year 2009 can be regarded as a period within which the pathogen developed rapidly. There were considerable differences in the intensity of infestation between individual research plots in 2009 (see the variance in the box plot in Figure 2). Later, damage intensity became uniform across all plots. On average, 60% of buds were infested on Colorado blue spruce during the monitoring period.

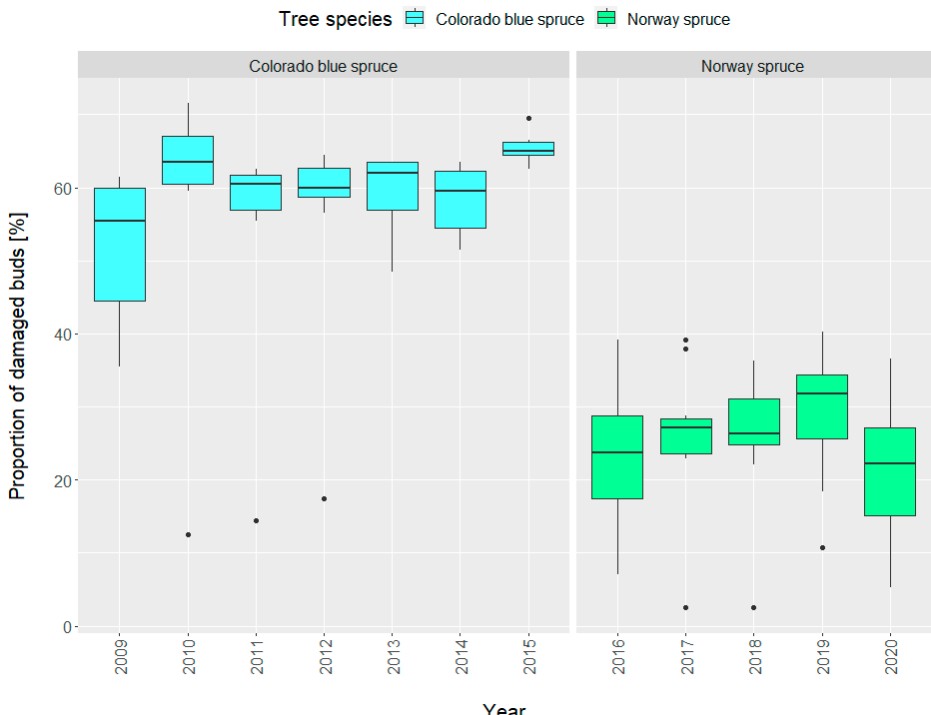

**Figure 2.** Proportion of buds infested by spruce bud blight in Colorado blue spruce (*Picea pungens* Engelm.) (2009–2015) and Norway spruce (*Picea abi*es (L.) Karst) in the age group 31–60 years. Horizontal line in middle of a box is the median, whiskers represent 1.5 times the interquartile range.

Only a few trees with individually infected buds were found in the stands of Norway spruce from 2009 to 2014 in the Ore Mountains region, even though the source of infection—infected buds of Colorado blue spruce—were in direct contact with them. In 2015, there was a significant increase in the damage to Norway spruce, and in many places entire stand groups of different ages were infested. In the research plots, an average 23% of buds were infested in 2016, the intensity was increasing and reached 31% by 2019, then declined to 22% in 2020 (Figure 2). In 2021, no new bud infestation could be found in the research plots, and a newly developed infection could be detected only in a few stand groups outside the research plots. During the main phase of pathogen development between 2016 and 2020, the intensity of bud damage in Norway spruce was around 25%.

In addition to the intensity of infestation of the tree canopy, the impact of spruce bud blight on Colorado blue spruce and Norway spruce also differed as to the extent of spatial infestation of the stands. In Colorado blue spruce, the distribution was continuous. All individual trees in the research plots were affected, as were those in the surrounding stands. Uninfected trees occurred only individually. Infestation was present to the comparable extent in both older and younger stands. The youngest stand infested by spruce bud blight was 11 years old. In the case of Norway spruce, the fungus was found in individuals younger than 15 years in only one stand, and the infestation of individual trees under 30 years of age was less frequent than in the case of Colorado blue spruce. In contrast to the Colorado blue spruce, there were also a number of Norway spruce stands in the Ore Mountains older than 60 years. The infestation of Norway spruce by spruce bud blight differed significantly depending upon stand age (GLMM: $n$ = 85; df = 12; $p$ < 0.01). In the initial (acute) phase of the pathogen's spread in 2016, an average of 80% of individuals in the 31–60 year age class were infested in the research plots, while only 42% of individuals were infested in older stands (Figure 3). The proportion of infested individuals changed during the period under study. There was a gradual decrease in the number of infested individuals in the age class of 31–60 years, and a sharp increase was recorded in the age class of 61–100 years in the final two years of the research period. In 2020, the proportion of

infested individuals in the 31–60 year age class was 68% and in the 61–100 year age class it was 62%. The relationship between the proportion of infested individuals and age of the stand varied during the monitored period. At the beginning of the survey in 2016, the proportion of infested individuals decreased with age, but this trend changed over time, due to both a drop in infestations in younger stands and an increase in infestations in older stands. In 2020, this trend more or less disappeared, but the comparison with 2016 was not statistically significant (GLMM: $n$ = 85; df = 12; $p$ = 0.06; Figure 4).

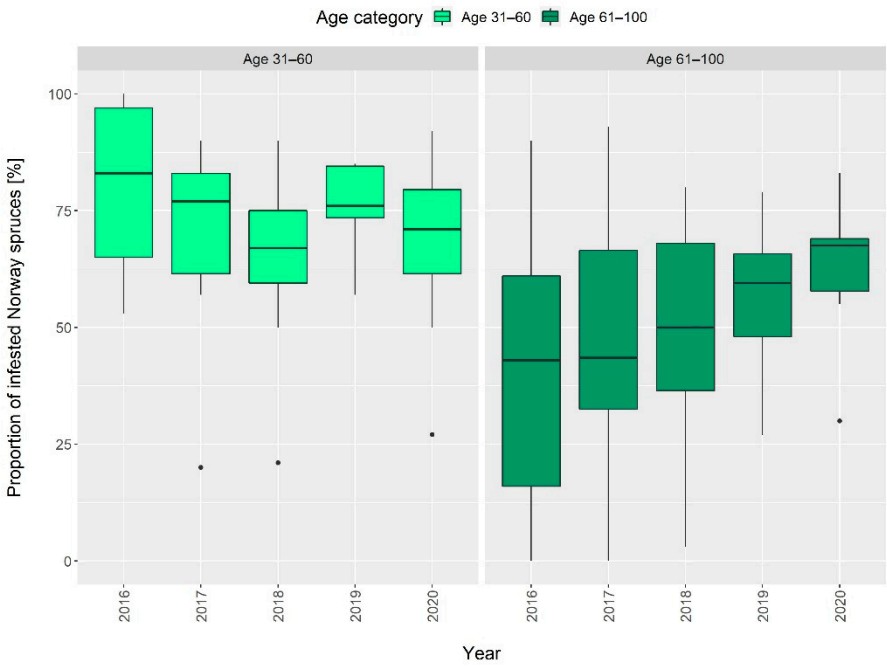

**Figure 3.** Proportion of individual Norway spruce trees infested by spruce bud blight by age classes in years 2016–2020. Horizontal line in middle of a box is the median, whiskers represent 1.5 times the interquartile range.

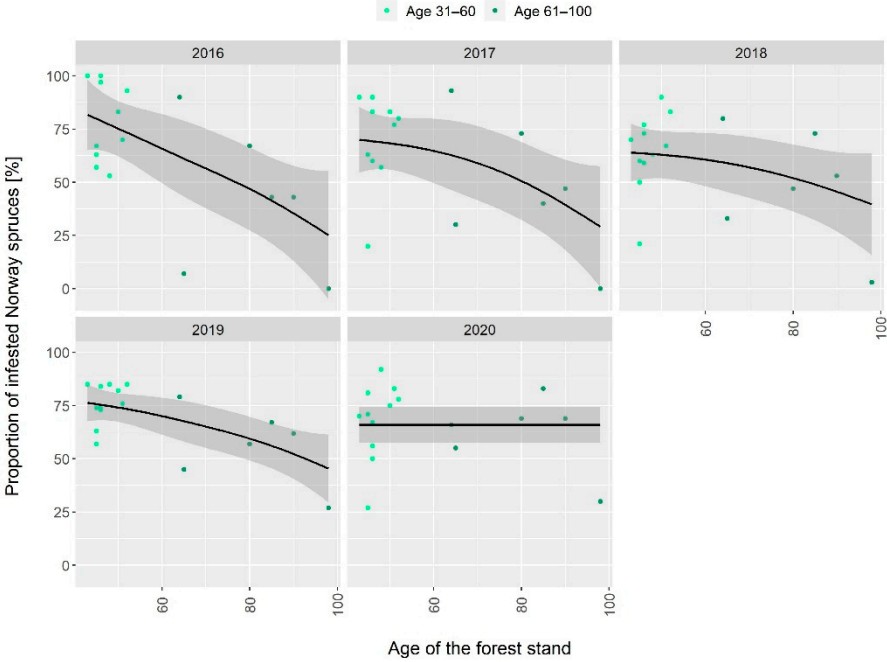

**Figure 4.** Comparison of the proportion of infested trees by age in years 2016–2020.

In terms of infestation severity in individual categories, there is a clear difference between younger and older stands of Norway spruce as to the proportions of heavily infested trees (categories 3 and 4), which was more than double in stands aged 31–60 years (Figures 5 and 6). The steady decline in the number of uninfected trees and gradual increase in infestation categories 1, 2, and 3 in the 61–100 year age class (Figure 7) is remarkable. By contrast, in the 31–60 years age class, there was a gradual regeneration of stands during 2016–2018, as represented by an increase in the number of uninfected trees, then a sudden deterioration of the situation in 2019, followed by another regeneration phase from 2020 onwards (Figure 5).

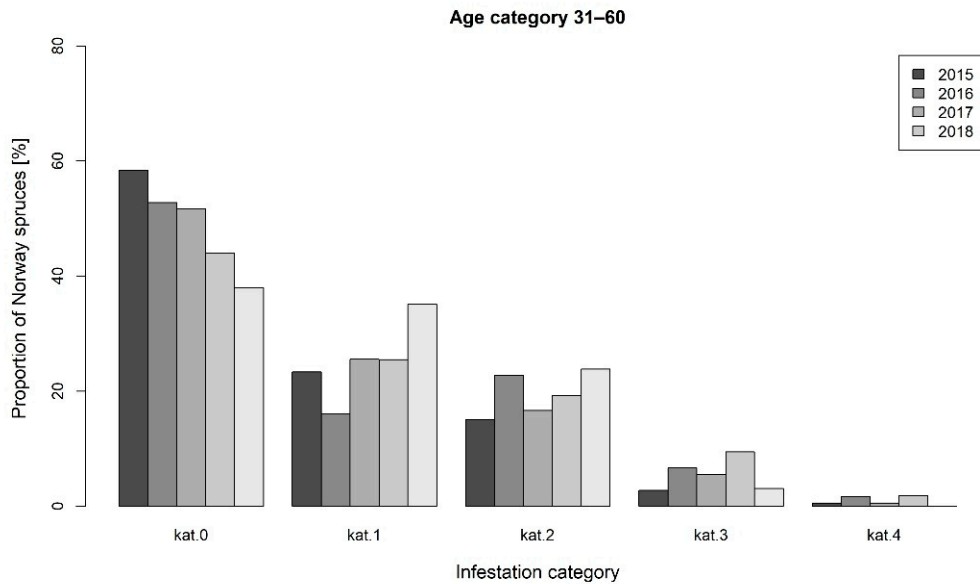

**Figure 5.** Proportions of infested Norway spruce trees by individual infestation categories in age class 31–60 years during 2016–2020.

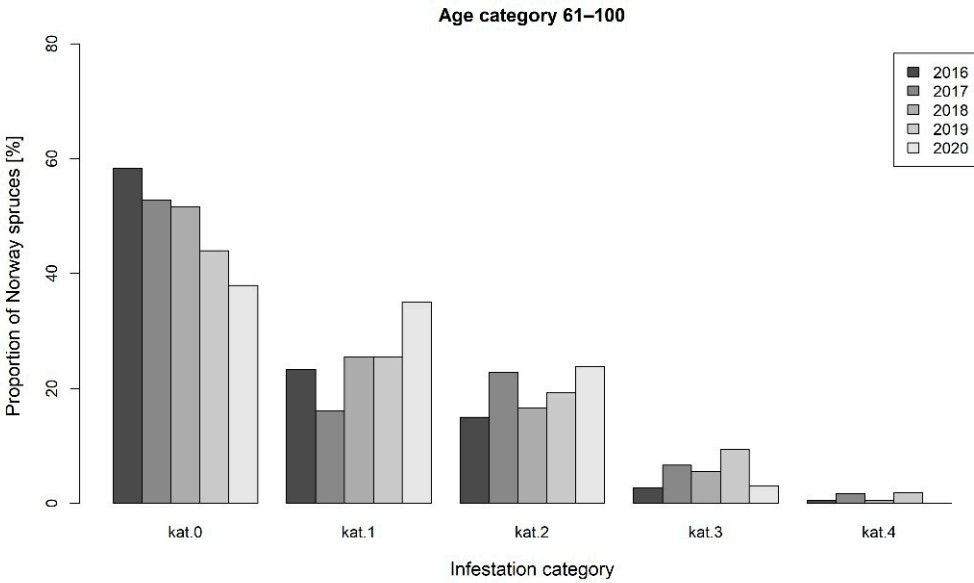

**Figure 6.** Proportion of infested Norway spruce trees by individual infestation categories in age class 61–100 during 2016–2020.

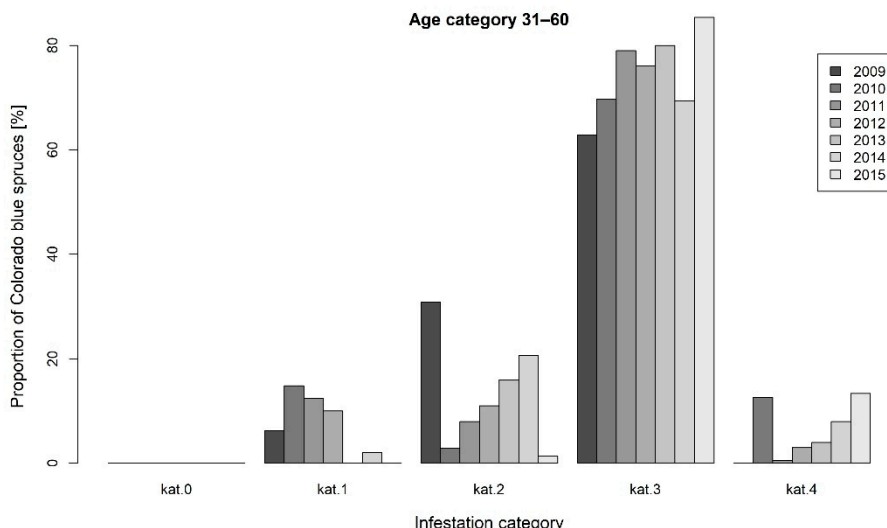

**Figure 7.** Proportions of infested Colorado blue spruce trees by individual infestation categories in age class 31–60 years during 2009–2015.

A completely different development of the pathogen can be observed in Colorado blue spruce in the age class of 31–60 years. This can be characterized as a continuous growth in infestation (Figure 6). The effect on Colorado blue was more pronounced by the pathogen, with an infestation category 3 (75% of individuals on average, Figure 6; for Norway spruce only 11% of individuals on average were in category 3, Figure 5). No uninfested trees were present in the Colorado blue spruce plots. The least damaged trees in category 1 were shifting to category 2, indicating a gradual deterioration of health (Figure 6).

Among the abiotic factors, damage to sprouting shoots with a subsequent infestation by the fungus *Botrytis cinerea* Pers., was observed in Colorado blue spruce. A frequent phenomenon wherein no damage to the buds occurred was dehydration (rusting of annual and, to a lesser extent, possibly older needles) during the dormant season and in very early spring, especially in sunny locations. Locally, ground-level ozone damage was investigated. None of the abiotic damage was of concern, and it was not very significant from the phytopathological point of view.

Among other fungal pathogens, *Lophodermium piceae* (Fuckel) Höhn. was observed on Colorado blue spruce and *Sirococcus conigenus* (Pers.) P.F. Cannon and Minter on shoots and needles. When they occurred heavily and repeatedly, both pathogens proved to be significant pests, and especially for young stands at sites with stable high air humidity. The genus *Armillaria* was frequently observed at the epicenter of occurrence on drying and dead individuals of both Colorado blue spruce and Norway spruce of various ages. Other fungi identified were of the genera *Rhizosphaera* and *Cytospora*, and some wood-destroying fungi (*Stereum sanguinolentum* (Alb. and Schwein.) Fr.) were found rather individually with no major significance for Colorado blue spruce or Norway spruce.

## 4. Discussion

The current dieback of Colorado blue spruce substitute stands, and deteriorated health of Norway spruce have their origins in the second half of the 20th century. Replacement stands were set up because of the significant damage to the original spruce cultures by air pollution, mainly by $SO_x$ and $NO_x$ deposition and high concentrations of hydrogen fluoride. Sulfur dioxide, due to its direct effect on the assimilation apparatus of trees and indirect effect on soil chemistry, subsequently caused vegetation to die in an area of approximately 40,000 ha [10]. Between 1990 and 2000, $SO_x$ emissions in the region of North Bohemia decreased by 87% and direct emissions have had practically no effect on the health status of vegetation today [10]. Nevertheless, the increased deposition of $NO_x$ and $NH_4^+$ ammonium ions continue in contributing to soil acidification. In these

acidic soils, low nutrient content is also a significant problem. Potassium content, for instance, is less than 3500 mg·kg$^{-1}$ in almost all parts of the Ore Mountains [44], which is a complication for growing trees in mountainous areas. The desulfurization of power plants that has been recorded in recent years [10,45,46] may be related to the current widespread spruce bud blight in the Ore Mountains. Sulfur actually acted as a fungistatic, thereby inhibiting development of the pathogen in these areas. When its content in pollutants was reduced, large scale fructification and spread of spruce bud blight occurred on Colorado blue spruce [20].

The negative impact of the aforementioned anthropogenic factors, resulting in biomass reduction, is evident [47]; furthermore, there has also been an increase in ozone (O$_3$) in recent years. Nevertheless, the persisting problem is the long-lasting soil acidification caused by air pollution. The poor soil fertility [44] reduces the defensive abilities of woody plants, resulting in the pathogen's successful attack [48]. Younger trees are generally more susceptible to air pollution than are mature trees, as they are able to obtain fewer nutrients needed for growth and securing of defense mechanisms [49]. Nevertheless, older stands can also be damaged by air pollution, especially because the canopy can act as a filter that reduces the amount of pollutants reaching the ground [50]. In addition, individuals in the understory generally have lower stomatal conductance, which results in less penetration by harmful substances into the tissue, thus contributing to less infestation of younger and shaded plants [51].

During the monitored period from 2009 to 2015, not a single completely healthy individual was observed in the stands of replacement trees composed of Colorado blue spruce. The infestation quickly reached disastrous proportions even before its peak in 2012. In 2013 and 2014, the observed infestation diminished slightly, but in 2015 the health of Colorado blue spruce deteriorated significantly. The infestation began to spread noticeably also on Norway spruce, even though, until 2015, and despite the significant impact of air pollution and decay of replacement stands of Colorado blue spruce, a major presence of spruce bud blight was not detected [22]. The situation abroad is similar, with only the following countries having confirmed the occurrence to date: the Czech Republic [20,40,52], Ireland [29], Italy [30], Alaska [53], and Great Britain [17]. No occurrence has yet been confirmed in Northern Europe. Pettersson [54] attributes this to low winter temperatures, which, despite the psychrophilic nature of the spruce bud blight, may limit the development of the pathogen.

The significantly poorer health of Colorado blue spruce caused by the pathogen can be attributed to two factors namely the choice of a tree species was not entirely appropriate for Czech Republic conditions because of different ecological valence andthe planting of Colorado blue spruce led to the destruction of the original ecosystems during reforestation and led to the removal of the original stands destroyed by pollutants [55]. The initially higher resistance to air pollution became completely irrelevant after desulfurization of power plants in 2000, especially as the fungistatic function of sulfur was eliminated and spruce bud blight spread as a result. Trees that replaced the original Norway spruces were unable to acclimate to Czech Republic conditions and, due to the presence of sulfur, did not develop the necessary resistance to the pathogen [56].

Although the highest level of infestation was found in stands aged 31–60 years, the trend of progressive infestation was most pronounced in the age category older than 60 years, mainly due to the increase in damaged individuals in categories 1 and 2. This difference was clearly visible in 2019 and 2020. To consider another example, in case of the pathogen *Hymenoscyphus fraxineus* (T. Kowalski) Baral, Queloz and Hosoya, more frequent infestations of younger ash trees were also observed [57]. Older trees tend to be more resistant to damage, but if the frequency of infection is higher in several successive years, then skeletal branches may start to drop off, which results in dying back. Younger ash trees are usually infected more frequently, mainly due to the spread of the pathogen from fallen leaves and petioles that are on the ground. In fact, intensity of spore dispersal on the ground is 5–100 times greater than at just 3 m above the ground [58]. Spruce bud blight



spreads primarily in the tree crown, where the number of buds may also play a role, and this is usually lower in younger individuals due to the denser canopy. Older individuals may initially be better able to resist pathogen infection due to their more developed defense mechanisms. If they are, however, exposed to infectious pressure repeatedly and unsuitable soil conditions for growth of woody plants persist over the long term [35], then the overall condition of trees may be worsened.

During the evaluation, other harmful factors of both an abiotic and biotic nature could also be observed, and to some extent these affected the overall health of the stands. In some years, frost damage to buds occurred frequently. In North America, spruce trees have been reported to be infested by the fungus *Dichomera gemmicola* A. Funk and B. Sutton, the bud blight from which may be confused with that of *G. piceae* at its first stage of development. The pathogen was previously observed 50 years ago in areas of eastern Canada [59]. Upon initial investigation, it may be confused on the basis of morphological traits and very similar symptoms, such as twisting of young shoots [35], but upon microscopic examination the fungi are easily distinguishable [60].

In recent decades, there has been an increasing trend in the average annual temperature and the number of days with a daily temperature of above 5 °C during winter months. Over the past 30 years, annual rainfall has been below the usual levels. This combination suggests an increase in drought stress during both winter and summer months, while the drought effect on the level of pathogen infestation has also been proven [61]. Thus, in the case of spruce bud blight, changes in climatic conditions may influence the intensity of infestation but are unlikely to be the most important factor at stand level. In the research plots, individuals with very different infestation rates were found almost side by side, and these individuals were equivalently influenced by climate and soil characteristics. From the physiological measurements, it is clear that the *G. piceae* negatively influences the water regime of the spruce and reduces the water-use efficiency ($WUE_{inst}$) in the shoots [62]. This can boosts the drought stress and suppress the resistance to further pathogen attack.

A similar and severe spread of pathogens has occurred in the past in the spruce stands of the Eagle Mountains in the Czech Republic [59], namely as a result of the fungus *Ascocalyx abietina* (Lagerb.) Schläpf.-Bernh., which is also known to have affected pine stands in Austria, Sweden, and Poland [63–66]. The pathogen appeared with similar dramatic suddenness and caused bud damage to both Colorado blue spruce and Norway spruce saplings. Years with deeper snow cover also played a role, creating optimal conditions for the pathogen to develop, thereby resulting in massive fructification, which caused its significant spread [63]. Weather conditions were also an important factor in its development, with the summer months having lower than average air temperatures and higher-than-average relative humidity. This established favorable conditions for the development of the pathogen and weakened trees' resistance. A similar phenomenon was recently observed in the Ore Mountains, where, after the colder year of 2017, a massive spread occurred in 2018. Provenance also influence the spruce's resistance. It is evident that lower altitude provenances tend to become more stressed by abiotic factors in mountain areas, making it easier for the pathogen to infect individuals. This is particularly true for *Pinus sylvestris* and *Pinus contorta* [65,67]. In the case of spruce, on the other hand, this has not yet been confirmed, although the negative effect of provenance suitability could also be observed [68]. To date, scientific verification as to the relationship between infestation intensity and the influence of spruce genotypes has not yet been carried out.

## 5. Conclusions

The main cause for the significant spread of bud-bling disease in the Ore Mountains since 2015 remains poorly understood. A complex set of causes is being considered, including unsuitable soil conditions (low pH and, typical of mountain ecosystems, low nutrient content, such as of potassium) [44]. In addition, one needs to consider the impossibility to predict climate fluctuations and, in the longer term, the development of soil conditions in mountainous areas, of which the influence has not yet been sufficiently described and quan-

tified in the context of pathogen infestation. Therefore, predicting further developments is complicated.

The influence of the surrounding strongly infected Colorado blue spruce trees also remains uncertain. If ideal conditions for the development of the fungus arise, then the breakdown of Norway spruce stands may occur, making spruce bud blight a key and conditional factor for spruce cultivation in the Ore Mountains. This hypothesis is based upon findings of the expanding range of the pathogen's infestation in Norway spruce. In 2018 and 2019, the occurrence of spruce bud blight on Norway spruce was confirmed in the area of the Jizera Mountains and Giant Mountains [69].

A combination of anthropogenic and biotic damaging factors could lead to the irreversible destruction of vegetation in mountain areas in the future. Moreover, trees weakened by air pollution or pathogen infection may become easy targets, especially for secondary harmful agents. Root rot could become a significant problem in the future, and especially *Armillaria* spp. [70], which, in combination with genera *Stereum* spp. and *Heterobasidion* spp., already pose major risks when growing woody plants on former agricultural lands, even in foothill areas. Following these threats, the cambiophagous insects, especially bark beetles (*Ips typographus* (L.), *Pityogenes chalcographus* (L.)) are in a state of permanent outbreak in the Czech Republic [71].

**Author Contributions:** Conceptualization, R.M., M.S., V.P.; methodology, V.P., R.M.; validation, M.S., V.P., R.M.; formal analysis, R.M.; investigation, M.S., V.P., R.M., J.V., D.B., I.T., F.L.; data curation, M.S., V.P., R.M., D.B., I.T., F.L., J.V.; writing—original draft preparation, M.S., V.P., R.M., I.T., F.L., D.B., J.V.; visualization, R.M., D.B., M.S., V.P.; supervision, R.M., V.P.; project administration, V.P., R.M.; funding acquisition, V.P., R.M. All authors have read and agreed to the published version of the manuscript.

**Funding:** This research was funded by GS LČR 68 under project "Picea abies attacked by Gemmamyces spruce bud blight—case study in the Ore Mountains"; by NAZV under the project "Objectivization of the method for detection of the occurrence and dynamics of forest damaging agents by modern remote sensing tools as a decision support mean for the state forest administrative" (grant number QK1920458), as provided by the Ministry of Agriculture of the Czech Republic. Infrastructural support and salary for R.M. and I.T. was also obtained from "EXTEMIT-K," No. CZ.02.1.01/0.0/0.0/15_003/0000433 financed by OP RDE. Infrastructural support and salary for V.P. was also obtained from "EVA4.0", No. CZ.02.1.01/0.0/0.0/16_019/0000803 financed by OP RDE.

**Institutional Review Board Statement:** Not applicable.

**Informed Consent Statement:** Not applicable.

**Data Availability Statement:** The data presented in this study are available on request from the corresponding author. The data are not publicly available due to policy of the institute.

**Acknowledgments:** The authors would like to thank: Aleš Kilb (forest manager district Litvínov, State Forest Enterprise Lesy ČR), who initialized damage research on the Colorado blue spruce in 2009; František Soukup, who assisted in establishing the first permanent plots in the Fláje area, and Daniel Tyšer who helped with the graphical abstract. The authors would like to thank Gale A. Kirking at English Editorial Services, s. r. o. for linguistic improvements.

**Conflicts of Interest:** The authors declare no conflict of interest.

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
