# Peer review of "Gemmamyces piceae Bud Blight Damage in Norway Spruce (Picea abies) and Colorado Blue Spruce (Picea pungens) Forest Stands"

_forests, doi:10.3390/f13020164_

Round 1
Reviewer 1 Report
The manuscript overall is good. The content of the paper could be published in Forests. In the manuscript, I pointed out a few minor corrections.

Author Response
The manuscript overall is good. The content of the paper could be published in Forests. In the manuscript, I pointed out a few minor corrections.
We accepted your changes in the manuscript and corrected the English language and style. We also reformulated some parts, which was sometimes misleading.

Reviewer 2 Report
In the manuscript presented by Sameket al. “Gemmamyces piceae bud blight damage in Norway spruce and Colorado blue spruce forest stands”. In this manuscript, the authors compare the development of bud blight infestation on the two host tree species. The study is significant and informative.
What is the main question addressed by the research? It summarizes present knowledge about the distribution, host tree species known to date, and describes the biology of the pathogen and factors influencing the spread of the fungus G. piceae.
Is it relevant and interesting? Yes, it is relevant.
How original is the topic? The authors compare the development of bud blight infestation on the two host tree species like Colorado blue spruce and Norway spruce, in terms of the intensity of damage to individual trees and the spatial extent of the infestation.
What does it add to the subject area compared with other published material? This paper summarizes the biology of G. piceae, its host plants and presents the first findings from the massive outbreak.
Is the paper well written? There is a need for some English language improvement in the manuscript.
Is the text clear and easy to read? Yes
Are the conclusions consistent with the evidence and arguments presented? Yes Do they address the main question posed? Yes
However, I found some faults, mistakes, or the leaving out of crucial details in the manuscript, which are discussed below.
- The authors should apply the statistical analysis for figures 5, 6 & 7.
- Add some recent references related to the article.
- All references should be in the same format. Please check it carefully.
- The authors should revise the manuscript accordingly and resubmit it.
Author Response
Is the paper well written? There is a need for some English language improvement in the manuscript.
We corrected and improved the English language and style. Native speaker Dr Gale A. Kirking (English Editorial Service) conducted the proofreading.
The authors should apply the statistical analysis for figures 5, 6& 7.
Figures 5, 6 & 7 complement the data analysed in previous parts. Differences between infestation categories and years are evident, and its testing would be rather self-serving if no other covariates were added (unfortunately are not available).
Further testing would also require a series of comparisons, and the results achieved may be less clear, moreover with no substantial improvement.
Add some recent references related to the article.
We added four new references (number 11, 12, 13 & 57). Generally, there is a considerably limited number of publications concerning the Gemammyces piceae.
All references should be in the same format. Please check it carefully.
We checked it and corrected it according to Reference List and Citations Style Guide for MDPI Journals.
